# Impact of Abiotic Stresses on Plant Virus Transmission by Aphids

**DOI:** 10.3390/v12020216

**Published:** 2020-02-14

**Authors:** Manuella van Munster

**Affiliations:** INRA, UMR385, CIRAD TA-A54K, Campus International de Baillarguet, CEDEX 05, 34398 Montpellier, France; manuella.van-munster@inra.fr; Tel.: +33(0)4-9962-4857

**Keywords:** Plant virus, abiotic stress, insect, vector transmission, drought, temperature, CO_2_, viral accumulation, aphid

## Abstract

Plants regularly encounter abiotic constraints, and plant response to stress has been a focus of research for decades. Given increasing global temperatures and elevated atmospheric CO_2_ levels and the occurrence of water stress episodes driven by climate change, plant biochemistry, in particular, plant defence responses, may be altered significantly. Environmental factors also have a wider impact, shaping viral transmission processes that rely on a complex set of interactions between, at least, the pathogen, the vector, and the host plant. This review considers how abiotic stresses influence the transmission and spread of plant viruses by aphid vectors, mainly through changes in host physiology status, and summarizes the latest findings in this research field. The direct effects of climate change and severe weather events that impact the feeding behaviour of insect vectors as well as the major traits (e.g., within-host accumulation, disease severity and transmission) of viral plant pathogens are discussed. Finally, the intrinsic capacity of viruses to react to environmental cues *in planta* and how this may influence viral transmission efficiency is summarized. The clear interaction between biotic (virus) and abiotic stresses is a risk that must be accounted for when modelling virus epidemiology under scenarios of climate change.

## 1. Introduction

Transmission is a key step in the life cycle of a virus, allowing viral maintenance in an ecosystem. Most plant viruses rely on a third party for host–host spread, with sap-sucking insects from the order *Hemiptera* (aphids, whiteflies, planthoppers, and leafhoppers) being by far the most widespread vectors [1,2]. Thus, virus transmission during the feeding process relies on a complex set of interactions that are the result of the co-evolution of at least three partners: the pathogen, the vector, and the host plant—each with its own behaviour, population, and community dynamics [3,4,5].

The feeding process is best described for aphids, but the same basic principles also apply to other species of the orders *Hemiptera* or *Thysanoptera*. Insect feeding can be monitored by the Electrical Penetration Graph (EPG) technique, which provides live visualization and recording of plant penetration by insect mouthparts [6,7]. Non-circulative transmission—in which the virus taken up by a vector on an infected plant attaches to the inner part of the cuticle lining of the feeding apparatus [8] and is subsequently released and inoculated into a new host plant—is the predominant strategy for plant virus–vector interactions. All three steps—acquisition, retention, and inoculation—generally occur within seconds to minutes and do not require passage within the vector’s body. In this system, viruses are acquired mainly during probing and transient puncturing of epidermal and mesophyll cells of infected leaf tissues (see [9] for review). The second category of viral transmission is designated circulative non-propagative transmission and is characterised by longer acquisition and inoculation periods (from hours to days), and long retention time in the vector body that can last several weeks, and often until the vector dies. In that case, the virus crosses the gut epithelium of the insect vector, diffuses into the haemolymph, and reaches and accumulates into the salivary glands without replicating. Finally, in the third category, designated circulative and propagative transmission, the virus completes a similar cycle within the vector’s body but replicates within the gut, the salivary glands, and sometimes other tissues of the insect before inoculation into a new host plant. These two latter modes of transmission are found mainly for phloem-limited viruses that are acquired and inoculated by insect vectors during long-lasting sap ingestion phases in sieve tubes (see [9] for review).

Many studies have provided evidence that viruses themselves might alter specific aspects of host plant phenotypes and/or vector behaviour in ways that enhance their transmission, also described as the “Vector Manipulation Hypothesis” [10,11,12,13,14]. Thus, plant viruses can modulate the amino acid composition of host sap [15,16]; plant defence pathways and their associated molecular signalling [17,18]; release of attractive volatiles [19,20,21]; and enhancement of yellowing disease symptoms on foliage [22,23], potentially favouring or disabling insect attraction, growth and reproduction, and finally colonisation of the host plant [17,24,25]. It is noted that in some cases, neutral effects of the virus infection on vector behaviour and performance might also be part of the adaptative strategy of the pathogen (see for review [13]).

Interestingly, convincing evidence that plant–insect interplay adapts in such a way that virus-induced changes in vector behaviour specifically match the viral mode of transmission has emerged from these studies (reviewed in [26,27]). Briefly, a virus transmitted in a circulative (either propagative or not) way should attract and arrest insect vectors, favouring settlement, reproduction, colony formation, and perhaps late production of “migrators”’ to ensure maximum dissemination. A non-circulative transmitted virus, in contrast, should have no effect, or should attract but then repel vectors, because its acquisition is fast and retention time extremely short [28,29,30].

## 2. Impact of Abiotic Stresses on Viral Transmission

Interactions between plants, viruses, and vectors are affected greatly by environmental conditions, ultimately shaping the spread of viral diseases (see [31] for review). Ongoing climate change is raising the frequency, intensity, and duration of abiotic stresses on plants, and climate prediction is pessimistic [32]. Among these changes, elevated temperatures, increased CO_2_ and O_3_ levels, and altered precipitation patterns are major stresses impacting plant performance, population dynamics, and ecology [33,34,35,36]. Under such conditions, and similarly to what happens upon virus infection, physiological changes in host plant status are also accompanied at the biochemical level by modification of sap composition, as well as alteration of phytohormones levels and signalling pathways [37,38], which may thus impact multiple plant virus transmission parameters (see [39,40] for review), as discussed below.

Temperature fluctuations have long been studied as a key environmental climate parameter shaping viral disease epidemiology [41,42,43]. The luteovirus *Potato leafroll virus* (PLRV) was shown to be better transmitted by *Myzus persicae* to *Physalis floridana* when it was both acquired and inoculated at a higher temperature (26 °C vs. 12 °C) [44]. This result is linked to previous work showing that a higher temperature reduces the latent period of PLRV within the vector, increasing the speed with which the virus moves from the gut to the salivary system via the hemolymph [45]. The optimum temperature corresponding to efficient disease development within the host in relation to the vector’s biology and transmission process (i.e., acquisition, latency, and inoculation phases) has been determined in several pathosystems. Studies on the transmission efficiency of *Banana bunchy top virus* by the aphid *Pentalonia nigronervosa* shows that both acquisition and inoculation of the virus are optimal at 25 °C. Temperature has a similar effect on the biology of *P. nigronervosa* with enhancement of vector fecundity at this temperature, suggesting that biological factors, such as aphid feeding behaviour, may have been of importance in determining the transmission rates [46]. It has been noted that the optimum temperature for plant virus transmission is pathosystem-dependent, and might differ slightly depending on the transmission step considered [47]. While the two potyviruses *Potato virus A* (PVA) and *Potato virus Y* (PVY) and PLRV are better transmitted, when acquisition by *M. persicae* is realised at 20 °C, the optimal temperature for further establishment of the infection differs according to virus species [48]. Optimal infection of PVA- or PVY- inoculated *N. benthamiana* occurs at 20 °C, while a temperature of 25 °C allows for the best disease development in PLRV-inoculated *P. floridana* [48].

Several studies have also focused on analyzing viral transmission efficiency under a combination of abiotic stresses including temperature [49,50,51]. In their study, Singh and colleagues looked at the inoculation step of PVY and PLRV under different conditions of temperature, relative humidity (RH), and light. While light intensity did not seem to affect virus infection, high temperature (25–30°C) and high RH (80–90%) increased transmission of both viruses by 30–35% [49]. Smyrnioudis and colleagues [50] highlight the role of the vector in the spread of the luteovirus *Barley yellow dwarf virus* (BYDV) under condition of drought and variable temperature. The positive effect of elevated temperature on aphid movement is exacerbated by the combination of drought and high temperature, significantly increasing BYDV transmission efficiency [50].

In recent years, there have been some reports on the effects of other abiotic factors, namely CO_2_, O_3_, and water stress, on the spread of viral diseases as a consequence of the extreme environmental changes [48,51,52,53,54,55,56,57,58]. Dader and colleagues [53] thoroughly examined the impact of elevated CO_2_ levels (eCO_2_) on *M. persicae* life history, feeding behaviour, and the virus transmission ability of *Cucumber mosaic virus* (CMV; cucumovirus) to pepper plants (*Capsicum annuum*), transmitted in a non-circulative manner. Potential effects of eCO_2_ on virus transmission and acquisition efficiency were evaluated under two CO_2_ regimes with several application timings. When both infected source plants and receptor plants had grown under eCO_2_ prior to transmission experiments, a two-fold decrease in CMV transmission was observed. The reason evoked to explain this decrease was the settlement of plant resistance mechanisms under eCO_2_ during virus inoculation rather than alterations in feeding behaviour of the vector. Indeed, EPG experiments did not reveal any differences in the number of stylet punctures that could have explained this alteration in transmission levels [53]. However, different results were obtained in PVY transmission experiments realised under similar conditions. When PVY-infected tobacco plants and *M. persicae* aphids were placed under eCO_2_ levels (800 ppm vs. 450 ppm), transmission efficiency increased [59]; however, no clear explanation for this increase was apparent.

The dramatic influence of a water deficit applied to *Brassica rapa* infected source plants on the transmission efficiency of two unrelated virus species (*Cauliflower mosaic virus*; CaMV; caulimovirus and *Turnip mosaic virus*; TuMV; potyvirus), transmitted in a non-circulative way by *M. persicae,* was reported [54]. In this latter study, alteration of the transmission process during the acquisition phase was assessed directly, and neither the aphids nor the receptor plants were submitted to any particular stress. Thus, the only parameter tested was the impact of drought on CaMV- or TuMV-infected *B. rapa* plants and the efficacy of virus acquisition depending on the water status of the infected plants. Under these conditions, a severe water deficit applied to CaMV- or TuMV-infected plants dramatically enhanced transmission by around 34% and 100%, respectively, without any change in within-host viral accumulation [54]. Although feeding behaviour was not analysed thoroughly by EPG monitoring, no distinct aphid behaviour that could explain the alteration in the transmission rate that was observed during the 2 min of feeding acquisition [54]. As discussed in the conclusions of this review, other factors linked to the intrinsic capacity of these two specific viruses to respond to changes in the physiological status of the host plant may be responsible for the observed increased transmission. Interestingly, transmission of *Soybean mosaic virus* (SMV), another potyvirus, was not impacted in a similar way by water deficit [57]. In that case, SMV transmission was reduced in soybean (*Glycine max* L.) plants under water deficit, while it increased significantly in a saturated water regime. It should be noted that in this latter report, both infected source plants, receptor plants, and vector populations were subjected to an identical water regime. Specific changes were observed in abscisic acid, salicylic acid, and jasmonic acid signalling according to the water stress condition (deficit vs. saturated) which may partly explain the impact on virus transmission and virus infection development [57].

## 3. Vector Feeding Behaviour under Abiotic Stresses

Several aspects of a vector’s life history (e.g., developmental time, longevity, fecundity, feeding habits, geographic distribution) are affected by climate change [60,61,62,63] and there have been recent attempts to predict long-term effects on insect dynamics [64,65]. In particular, it is well known that aphid vectors react strongly to small changes in temperature or eCO_2_ due to their short generation times and great capacity for reproduction [66].

In particular, as evoked in the previous section, abiotic stresses may affect the feeding behaviour and fitness of sap-sucking insects through changes in plant physiology, alteration of host plant quality (i.e., secondary metabolites, essential amino acids, carbohydrates pools, etc.), and activation of plant defences [67,68,69], with potential consequences for virus transmission efficiency [53,54,56]. Some aphid species are reported to have a greater reproductive capacity under drought conditions than under non-drought conditions [70]. The most accepted explanation for this phenomenon was proposed by White and colleagues [71] in their study on psyllids—closely related to aphids—who suggested that drought increases the hydroxylation of proteins, which subsequently increases the levels of free amino acids available. Analysis of aphid feeding behaviour also indicates that drought stress increases mesophyll/phloem resistance [72], plausibly due to changes in phloem sap viscosity as a result of altered sugar and solute concentrations, making it harder for aphids to acquire nutrients [73]. Changes in the water potential of the host plant due to water stress can also increase the aphids’ ability to consume xylem sap, allowing aphids to deal with high sugar concentrations and osmotic pressure of the phloem sap [72,74]. EPG experiments conducted on *Aphis glycines* showed that feeding behaviour is significantly altered by different water stress regimes [57]. Aphids showed a significantly greater number of stylet punctures when reared on soybean plants grown under a saturated water regime compared to a drought treatment. In terms of behaviour critical for non-circulative transmission, this feature could explain the higher transmission rate of SMV reported in this study [57]. Interestingly, the authors also showed that irrespective of the water stress treatment, non viruliferous aphids show a greater number of stylet punctures compared to viruliferous aphids, highlighting the influence of the virus on the feeding behaviour and thus on the outcome of the infection. In another study, when a water deficit was applied to *Arabidopsis thaliana* (Col-0) infected with *Turnip yellows virus* (TuYV), a phloem-restricted luteovirus, transmission efficiency was reduced by 50%, most probably through alteration of feeding behaviour of the vector [55]. Indeed, while TuYV within-host accumulation was not altered by drought, virus load was reduced drastically in *M. persicae* vectors feeding on water-stressed TuYV-infected *A. thaliana* plants [55]. This result suggests an increase of aphid feeding from xylem vessels to balance the osmotic pressure of the sugar-rich phloem sap and avoid dehydration [75,76].

Several studies have considered the vectors life’s history during the tripartite interaction virus–vector–host plant under eCO_2_ [53,77,78]. As generally reported, the eCO_2_ concentration broadly affects plant physiology in both a positive and a negative manner. Positive effects of eCO_2_ include increased plant height and aboveground biomass, increased rates of photosynthesis, greater light use efficiency, and higher water-use efficiency due to the partial closure of stomata [79,80,81]. However, eCO_2_ can cause serious changes to plant biochemistry, including an increase in the carbon (C) to nitrogen (N) ratio caused by a reduction in foliar N and an increase in C due to higher growth and photosynthetic rates [80]. Although overall trends among insects reared on plants exposed to eCO_2_ show either a negative effect on population growth or increased feeding rates to compensate for lower N content [82,83], aphids are one of the few examples where eCO_2_ can increase population abundance, but this is often species and host specific [84]. Concerning aphid feeding behaviour, decreased salivation into sieve elements, increased phloem sap ingestion and a reduction of penetration attempts are among the responses observed for *Acyrthosiphon pisum* on *Medicago truncatula,* and *M. persicae* on pepper plants under eCO_2_ to maintain the quality of ingested nutrients [53,63]. The same feeding behaviour pattern was observed for *Rhopalosiphum padi* feeding on noninfected wheat plants grown under eCO_2_, with a 34% increase in the duration of phloem ingestion compared with plants grown under normal conditions [77]. It is believed that the objective of this altered feeding behaviour is to compensate for the significant decrease in N content in plants growing under eCO_2_. More surprisingly, BYDV infection is able to mediate the effects of eCO_2_ on wheat by partially restoring N content, consequently re-establishing vector performance and feeding behaviour [77].

## 4. Alteration of Viral Traits under Abiotic Stresses

The perturbating effects of abiotic stresses on host physiological status might also influence the life cycle of viruses and the relationships between viral traits such as within-host accumulation and disease severity (or virulence), introducing the possibility of an additional layer of complexity that may have an impact on virus transmission itself [39,40,85].

Plant signalling pathways and responses to various abiotic stresses are partly shared with those induced by viral infection, and the fact that they can interfere with one another is not a novel concept (see [86] and references therein). The effect of abiotic/biotic plant stresses on viral accumulation through the hijacking of plant signalling and defence pathways has received much recent attention [86,87,88,89]. Plant abiotic stress sensing, likely through the Ca^2+^ signalling pathway, was shown for the potyvirus PVA [86]. When PVA-infected *N. benthamiana* leaves were subjected to salt, osmotic, or wounding stress, the PVA gene expression increased, probably due to an elevation in cytosolic Ca^2+^ concentration. Similarly, BYDV, one of the most widely distributed viral diseases of cereals, significantly increases its titre by more than 30% in wheat growing under eCO_2_ or elevated temperature [87,90].

However, this positive effect of eCO_2_ concentration on virus accumulation cannot be generalized. In other pathosystems, namely *Tomato yellow leaf curl virus* (TYLCV), a geminivirus transmitted by the whitefly *Bemisia tabacci*/*Solanum lycopersicum* (tomato) and PVY/*Nicotiana tabacum* (tobacco), virus accumulation decreased significantly under eCO_2_ levels [91,92,93]. In both these latter cases, this observation was correlated with a decrease in disease severity due to the modulation of phytohormones by eCO_2_ levels. In PVY, a combination of elevated temperature and CO_2_ levels [i.e., 30 °C and 970 parts-per-million (ppm), respectively] decreased viral titres in *N. benthamiana* plants drastically compared with standard conditions (25 °C, ~405 ppm CO_2_), ultimately reducing the probability of transmission by *M. persicae* [51]. In the case of SMV, accumulation was reduced in soybean plants experiencing a severe water deficit, while viral infection increased significantly under water-saturated conditions [57]. Similarly, CaMV accumulation was reduced in several *A. thaliana* accessions subjected to water deficit [52]. Moreover, under these conditions, disease appearance and development were negatively affected, likely due to a slower systemic movement and the long-distance transport of viral particles through phloem [52,94].

Several factors, such as more efficient RNA silencing-mediated plant defence, may explain the lower incidence of virus disease under abiotic stress [95,96,97]. Chung and colleagues [98] examined the effects of various temperatures on TuMV symptom intensity and the speed of systemic infection in *B. campestris*. Plants mechanically inoculated with TuMV and subjected to a temperature gradient (from 13° to 33 °C) showed a complete loss of viral symptoms associated with lower virus accumulation at higher temperatures. The authors assumed that the higher temperature prohibited virus replication and movement due to the activation of plant defence mechanisms [85,99]. Similarly, viral accumulation of the cucumovirus *Peanut stunt virus* showed a rapid increase at the beginning of the infection at a higher temperature (27 °C vs. 21 °C), followed by a dramatic decrease in *N. benthamiana* plants due to the induction of plant defences [100].

However, it seems that the strength of antiviral silencing at a high temperature, or those of the viral suppressors that counteract it, may not be the only determinant, and is pathosystem specific [101]. CMV accumulation within *N. benthamiana* was not affected when plants were grown at 30 °C, while PVY accumulation was strongly reduced by 50% compared with standard growing conditions (25 °C) [101]. Interestingly, P23, a viral suppressor of RNA silencing of the crinivirus *Lettuce chlorosis virus*, induced local necrosis in *N. benthamiana* plants with increased severity at raised temperatures [102]. This observation is intriguing since it is generally reported that an increase in necrosis severity is enhanced at lower temperatures for viruses with a known efficient RNA silencing suppressor [95,103,104].

## 5. Conclusions

Changes to plant distribution, growth rates, plant physiological status, and biochemistry mediated by future climates are likely to have a major impact on insect vector biology, feeding behaviour and, ultimately, the spread of viral disease. The unequivocal conclusion is that abiotic stresses (e.g., temperature, CO_2_, O_3_, water stress) can have a dramatic effect on viral transmission rates [48,53,54,55,57].

In particular, there is direct evidence that infected plants subjected to drought can facilitate viral transmission and that this effect is independent of other putative factors such as the attraction of vectors and modification of their fitness [54]. Although the underlying mechanisms remain elusive at this stage, it has been suggested that the dramatic increases in CaMV and TuMV transmission rate are due to changes in the host plant physiological status that could trigger a direct effect on virus behaviour [105]. To date, this new concept, called “perceptive viral behaviour”, has been shown for a few plant viruses transmitted non-persistently, namely CaMV, TuMV and PVY, and describes the notion that viruses can “sense” aphid feeding activity, as well as some abiotic stresses, and immediately and reversibly produce transmissible morphs or provoke a viral cellular relocalization [106,107,108]. Such rapid viral reaction actually predisposes the infected plant to more efficient virus acquisition and transmission by aphid vectors. This remarkable phenomenon has been designated “transmission activation” and it can be triggered by abiotic stresses such as CO_2_ treatments [109].

Moreover, while beyond the scope of the present review—although it highlights the entanglement of plant–virus interactions—is the observation of the protective effect of virus infection on plant responses to abiotic stresses. This phenomenon has been observed for several viruses (CMV, *Brome mosaic virus*, TMV, *Tobacco rattle virus*, CaMV and TuMV) and a large panel of host plants, including rice (*Oryza sativa*), beet (*Beta vulgaris*), tobacco, *N. benthamiana*, tomato, *Solanum habrochaities*, pepper, watermelon (*Cucumis lanatus*), cucumber (*Cucumis sativus*), zucchini squash (*Cucurbita pepo*), *Chenopodium amaranticolor*, *A. thaliana*, barley (*Hordeum vulgare*) and *B. rapa* [54,110,111,112]. These findings suggest that host responses to virus infection that engender drought or cold resistance, such as alterations in small RNA pathways, responses to abscisic acid, changes in the metabolism of osmoprotective compounds, or effects on salicylic acid-mediated signalling are widely conserved [110,111,112,113]. They also support the concept that viruses can, at least under certain conditions, act as mutualistic symbionts [114].

Together with the consistent observation that several virus species react to environmental changes in ways that lead to an increase in transmission, a situation where infected plants can better survive adverse environmental conditions and are better sources for viral transmission, invite further investigation. Indeed, a clear interaction between biotic (virus) and abiotic stresses is seen as a risk that must be accounted for when modelling virus epidemiology under scenarios of climate change.

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
