# Peer review of "Impact of Abiotic Stresses on Plant Virus Transmission by Aphids"

_viruses, 2020, doi:10.3390/v12020216_

Round 1

Reviewer 1 Report

The review by van Munster provides an exhaustive overview of the impacts of different abiotic stresses on plant-virus-vector interactions, focusing specifically on viral transmission, viral traits and vector behavior. Overall, the author provides a nice review of the current knowledge and emphasizes the importance of studying viral transmission within the context of climate change/global warming. The review is well-written, timely and worthy of publication. I have a few suggestions and minor comments that I think would improve the quality of paper:

The author provided numerous examples about impacts of temperature fluctuations, water stresses and elevated CO2 but I was also expecting some information about O3 (ozone) effects. I think fewer studies have been conducted on this topic. Nevertheless, it might be interesting to mention it quickly and perhaps include it in perspective.

In the last section of the introduction, the author describes what is called “manipulation hypothesis” or “parasite manipulation” without mentioning it. Please, consider adding this precision if you think it's suitable.

In the same section, the predictions made by the author are a little too “strong” / “categorical”. Indeed, reviews on the subject point out that neutral effects on vector behaviors/performances might also be adaptative for the viruses (e.g. Eigenbrode et al. 2018 or Mauck et al. 2018).

L.60. To my knowledge, recent virus classifications include propagative viruses as circulative viruses. Thus, viruses are not propagative or circulative. I would adjust the sentence to “transmitted in a circulative (either propagative or not) way”.

L.84-86. How can the “enhancement of vector fecundity at this temperature” be responsible for a variation in acquisition/inoculation of the virus? Is it because of an increased population dynamic? Something physiological? Please, give some more information. In the same sentence “both acquisition and inoculation of the virus were”.

L.140-146. Consider moving this example (Arabidopsis – TuYV) in the next section as the drought effect on transmission seems mostly linked to vector behavior alteration.

L.186. What does “a shorter nonpathway phase” mean (xylem, derailment, phloem phases?)? Can you please be more specific?

Author Response

Dear Editor,

I would like to thank the Reviewers for their detailed assessment of my manuscript. Their comments and suggestions significantly helped its improvement.

Below you will find answers to the specific comments of the reviewers (blue indented text). References to lines LXXX are to line number in the main text without track changes.

Reviewer 1

The review by van Munster provides an exhaustive overview of the impacts of different abiotic stresses on plant-virus-vector interactions, focusing specifically on viral transmission, viral traits and vector behavior. Overall, the author provides a nice review of the current knowledge and emphasizes the importance of studying viral transmission within the context of climate change/global warming. The review is well-written, timely and worthy of publication. I have a few suggestions and minor comments that I think would improve the quality of paper:

The author provided numerous examples about impacts of temperature fluctuations, water stresses and elevated CO2 but I was also expecting some information about O3 (ozone) effects. I think fewer studies have been conducted on this topic. Nevertheless, it might be interesting to mention it quickly and perhaps include it in perspective.

Precisions on the effect of O3 on plant performance and viral transmission were added line 120, and line 196 as well as references on this subject.

In the last section of the introduction, the author describes what is called “manipulation hypothesis” or “parasite manipulation” without mentioning it. Please, consider adding this precision if you think it's suitable.

The term “Vector Manipulation Hypothesis’ was added line 71, referring to Ingwell et al., 2012.

In the same section, the predictions made by the author are a little too “strong” / “categorical”. Indeed, reviews on the subject point out that neutral effects on vector behaviors/performances might also be adaptative for the viruses (e.g. Eigenbrode et al. 2018 or Mauck et al. 2018).

As suggested by the reviewer, one sentence was added (line 77) to precise this point and refers to Mauck et al. 2018.

L.60. To my knowledge, recent virus classifications include propagative viruses as circulative viruses. Thus, viruses are not propagative or circulative. I would adjust the sentence to “transmitted in a circulative (either propagative or not) way”.

Line 80 : added

L.84-86. How can the “enhancement of vector fecundity at this temperature” be responsible for a variation in acquisition/inoculation of the virus? Is it because of an increased population dynamic? Something physiological? Please, give some more information. In the same sentence “both acquisition and inoculation of the virus were”.

This is a good remark of the reviewer. I have rephrased and completed the sentence to avoid confusion (Line 136).

“both acquisition and inoculation of the virus were” : fixed

L.140-146. Consider moving this example (Arabidopsis – TuYV) in the next section as the drought effect on transmission seems mostly linked to vector behavior alteration.

As suggested by the reviewer, the example (Arabidopsis – TuYV) was moved in the next section.

L.186. What does “a shorter nonpathway phase” mean (xylem, derailment, phloem phases?)? Can you please be more specific?

This is a good remark of the reviewer and I now precise (Line 321) the meaning of “nonpathway phase” in the manuscript. It refers to a reduction of penetration attempts under eCO2 linked to an increase of passive feeding, a strategy to maintain the quality of ingested nutrients (see Guo et al., 2013).

Reviewer 2 Report

General comments:

This is a well-written and timely contribution on an important topic in virus epidemiology. HoweverI have a few minor concerns: Firstly, the title is misleading - the paper is more than a review on effects of abiotic stress on aphid transmission. Secondly, and as a consequence, some of the sections in the paper appear out of scope as noted in the specific comments below. The solution in my view is to rework the title, abstract and Introduction to better reflect the content ("transmission" of course should still appear in the title and as a major emphasis) and to make some minor restructuring adjustments to the text.

Specific comments:

line 29. What is meant by "the most efficient vectors"? Do you mean the most widespread or pervasive. Even for aphids there can be huge differences between species in acquisition or inoculation efficiency.

line 49. inoculation into (or just to)

lines 68-69. "Earth's climate prediction is pessimistic" needs clarifying rephrasing. Presumably you mean the predictions for human society?

line 159. Delete "closely related to aphids". I don't think the readership of this journal don't need being told this information.

lines 175-194. The section on the link between plant physiology and vector performance and feeding behaviour is well documented and an important part of the paper.

lines 200-210. I'm unsure of the relevance of these TMV studies. Do they really indicate the "opportunities for transmission"?

lines 230-248. Again, although the relationship between RNA silencing, abiotic stress and plant defence mechanisms is well documented, I'm unsure of the relevance for a review paper on aphid transmission.

lines 267-278. Rather than having this as a paragraph in the Conclusions, I suggest there should be a section in the main text with an appropriate heading on protective effects of plant viruses against abiotic stress.

Author Response

Reviewer 2

This is a well-written and timely contribution on an important topic in virus epidemiology. However I have a few minor concerns: Firstly, the title is misleading - the paper is more than a review on effects of abiotic stress on aphid transmission.

Secondly, and as a consequence, some of the sections in the paper appear out of scope as noted in the specific comments below. The solution in my view is to rework the title, abstract and Introduction to better reflect the content ("transmission" of course should still appear in the title and as a major emphasis) and to make some minor restructuring adjustments to the text.

To answer to this major suggestion of the reviewer I have accordingly changed the tittle of this review in order to better reflect the content of the paper. However, I do not think that the main text needs to be adjusted since this new title reflects the different topics developed.

Specific comments:

line 29. What is meant by "the most efficient vectors"? Do you mean the most widespread or pervasive. Even for aphids there can be huge differences between species in acquisition or inoculation efficiency.

The comment of the reviewer is relevant and the term “efficient” was changed to “widespread” (Line 46)

line 49. inoculation into (or just to)

Fixed

lines 68-69. "Earth's climate prediction is pessimistic" needs clarifying rephrasing. Presumably you mean the predictions for human society?

I rephrased the sentence to clarify its meaning (line 119).

line 159. Delete "closely related to aphids". I don't think the readership of this journal don't need being told this information.

I do not agree with this comment of the reviewer. As said in this paragraph, aphid fecundity may be positively influenced by drought conditions. The most probable explanation is given by a study done on another Hemiptera family. Thus, I think this is important to mention that psyllids belong to a closely related family hence strengthen the comparison with aphids.

lines 175-194. The section on the link between plant physiology and vector performance and feeding behaviour is well documented and an important part of the paper.

lines 200-210. I'm unsure of the relevance of these TMV studies. Do they really indicate the "opportunities for transmission"?

This remark of the reviewer is highly relevant. Indeed, there is no clear conclusions on the effect of methanol vapours on transmission efficiency. Moreover, Sacristan et al. have shown that an increase of TMV accumulation did not influence transmission efficiency. Thus, I removed the description of TMV study from the manuscript.

lines 230-248. Again, although the relationship between RNA silencing, abiotic stress and plant defence mechanisms is well documented, I'm unsure of the relevance for a review paper on aphid transmission.

As evoked by the reviewer, this review paper extends beyond vectored transmission, and major virus traits (e.g. virulence and accumulation) are also discussed. In that context, how abiotic stresses can modulate RNA silencing and virulence seems to me important to discuss.

lines 267-278. Rather than having this as a paragraph in the Conclusions, I suggest there should be a section in the main text with an appropriate heading on protective effects of plant viruses against abiotic stress.

Concerning this remark of the reviewer, even though the protective effect of viruses under abiotic stresses is a really interesting topic I do not think it would be appropriate to put it in the main text (and it is the reason I mentioned “while beyond the scope of the present review” Line 457). This review means to present how virus traits and aphid feeding behaviour can be impacted by abiotic stresses and how it may influence virus transmission.

By discussing this feature in the Conclusion it may open the discussion on the impact of abiotic stresses on virus infected-plants and how it may interfere with virus spread.